# Relation of Brain Perfusion Patterns to Sudden Unexpected Death Risk Stratification: A Study in Drug Resistant Focal Epilepsy

**DOI:** 10.3390/bs12070207

**Published:** 2022-06-24

**Authors:** Lilia Morales Chacon, Lidice Galan Garcia, Jorge Bosch-Bayard, Karla Batista García-Ramo, Margarita Minou Báez Martin, Maydelin Alfonso Alfonso, Sheyla Berrillo Batista, Tania de la Paz Bermudez, Judith González González, Abel Sánchez Coroneux

**Affiliations:** 1International Center for Neurological Restoration, 25th Ave, No 15805, Playa, Havana PC 11300, Cuba; kbatista.gr@gmail.com (K.B.G.-R.); minou@infomed.sld.cu (M.M.B.M.); alfonsomaydelin@gmail.com (M.A.A.); sheylabb@infomed.sld.cu (S.B.B.); taniadelapazbermudez@gmail.com (T.d.l.P.B.); judith@neuro.ciren.cu (J.G.G.); abel@neuro.ciren.cu (A.S.C.); 2Cuban Neurosciences Center, 25th Ave, No 15202, Playa, Havana PC 11300, Cuba; lidicegalan2000@yahoo.com; 3McGill Centre for Integrative Neuroscience, Ludmer Centre for Neuroinformatics and Mental Health, Montreal Neurological Institute, Montreal, QC H3A 0G4, Canada; oldgandalf@gmail.com

**Keywords:** drug resistant focal epilepsy, SUDEP, biomarkers, single photon emission computed tomography, ictal SPECT, interictal SPECT

## Abstract

To explore the role of the interictal and ictal SPECT to identity functional neuroimaging biomarkers for SUDEP risk stratification in patients with drug-resistant focal epilepsy (DRFE). Twenty-nine interictal-ictal Single photon emission computed tomography (SPECT) scans were obtained from nine DRFE patients. A methodology for the relative quantification of cerebral blood flow of 74 cortical and sub-cortical structures was employed. The optimal number of clusters (K) was estimated using a modified v-fold cross-validation for the use of K means algorithm. The two regions of interest (ROIs) that represent the hypoperfused and hyperperfused areas were identified. To select the structures related to the SUDEP-7 inventory score, a data mining method that computes an automatic feature selection was used. During the interictal and ictal state, the hyperperfused ROIs in the largest part of patients were the bilateral rectus gyrus, putamen as well as globus pallidus ipsilateral to the seizure onset zone. The hypoperfused ROIs included the red nucleus, substantia nigra, medulla, and entorhinal area. The findings indicated that the nearly invariability in the perfusion pattern during the interictal to ictal transition observed in the ipsi-lateral putamen F = 12.60, *p* = 0.03, entorhinal area F = 25.80, *p* = 0.01, and temporal middle gyrus F = 12.60, *p* = 0.03 is a potential biomarker of SUDEP risk. The results presented in this paper allowed identifying hypo- and hyperperfused brain regions during the ictal and interictal state potentially related to SUDEP risk stratification.

## 1. Introduction

Drug-Resistant Focal Epilepsy (DRFE) has been related to unfavorable treatment outcomes, lower life quality, higher risk of seizure-related injuries, and increased risk of sudden unexpected death (SUDEP). Although cardio-respiratory brainstem dysfunction induced by bilateral tonic clonic seizures appears to be the most likely cause, the exact pathophysiology of SUDEP is still unknown [1,2].

The risk of SUDEP upsurges with the frequency and severity of uncontrolled seizures. Revised SUDEP-7 Inventory (range: 0–10) has been applied to gauge physiologic and diagnostic biomarkers for SUDEP in many studies, revealing association with heart rate variability, postictal EEG suppression, and neuroimaging studies as potential biomarkers [3,4,5,6].

Numerous advances in structural and functional neuroimaging and computational neuroscience suggest that neural dysfunction in focal epilepsy is not restricted to the epileptogenic zone (EZ) but can affect widespread networks throughout the brain. These findings may lead to improved strategies for outcome prediction in epilepsy and EZ localization [7,8,9,10].

Structural and functional neuroimaging studies such as electroencephalogram-gram (EEG), magnetic resonance imaging (MRI), single-photon emission computed tomography (SPECT), and magnetoencephalography have made it possible to characterize some alterations of the brain in individuals who are at high risk as well as in those who succumb to SUDEP [11,12,13,14]. Improved understanding of the specific networks underlying human interictal and ictal states may also help to elucidate targets for new therapy to diminish SUDEP risk.

Moreover, studies with functional resting-state show reduced functional connectivity in the anterior cingulate cortex–thalamus–brainstem network in patients at high risk of SUDEP [15,16]. Reported findings encompass alterations in the volume and thickness of cortical and subcortical structures that intervene in autonomic regulation [17].

Even though it has been recognized that innovative brain-imaging modalities tracking neurovascular dynamics during seizures may provide new disease biomarkers, we only found two studies of perfusion neuroimages associated with SUDEP in the scientific literature to date. One of these studies, which was carried out in patients with focal epilepsy who underwent perfusion imaging with arterial spin labeling magnetic resonance imaging (MRI) determined that postictal hypoperfusion in brainstem respiratory centers occurs more frequently following bilateral tonic–clonic seizures than other seizure types, providing a possible explanation for the increased risk of SUDEP in patients who recurrently experience this kind of seizure [1]. The other study, involving SPECT revealed that abnormal increased activity in subcortical structures as well as a decreased activity in the association cortex may be vital for clinical manifestations in tonic–clonic seizures, which provide a possible explanation for the increased SUDEP risk [18]. SPECT is extensively used to study neuronal activity due to the coupling between blood flow and cerebral metabolism [19]. Although the relationship between SPECT and surgical outcome has not been established [20], multimodal imaging using SPECT coregistered to MRI are useful to improve EZ localization. The concordance of these techniques with other tests such as EEG, MRI, and semiology are predictive for good seizure outcome after epilepsy surgery [9,21,22]. In addition, we have reported that brain perfusion patterns in patients with DRFE show hyperperfusion in brain structures related to the EZ [23].

Imaging studies have contributed to the analysis of functional connectivity between cortical and subcortical autonomic regulatory brain structures. There is also evidence that altered structural and functional connectivity between brain regions are associated with SUDEP [14,16,24]. Even though some investigations on MRI suggest that it may be possible to predict progression to SUDEP, combinations with additional neuroimaging modalities can offer complementary information [25,26]. Here, we used a methodology for the relative quantification of cerebral blood flow from the coregistration of functional (SPECT) and structural images (MRI) to explore the role of the interictal and ictal SPECT to identity neurofunctional imaging biomarkers for SUDEP risk stratification in subjects with DRFE.

## 2. Materials and Methods

In the present study, 29 interictal-ictal SPECT scan pairs were obtained from nine adult patients with DRFE. Clinical and demographic information was tabulated. Diagnosis was determined according to the International Center for Neurological Restoration Comprehensive Epilepsy Surgery Program, which included (a) prolonged video-electroencephalography (VEEG)monitoring with scalp electrodes and additional electrodes considering the epileptogenic zone presumed; (b) MRI scans with a 1.5 T or 3 T scanner (Siemens Magnetom Symphony); (c) neuropsychological testing (executive functions, attention and memory assessment, higher verbal and visual functions); (d) sensorial evoked potentials, (e) antiepileptic drugs blood levels determination; and (f) subtraction single-photon emission computed tomography coregistered to magnetic resonance imaging (MRI) (SISCOM) [9,27]. Patients were prospectively recruited for this study from December 2016 to February 2020 after providing informed consent.

All subjects were gauged with SUDEP-7 Risk Inventory (SUDEP-7), including seven validated and weighted risk factors firstly identified by Walczak et al. as factors associated with SUDEP risk [28]. SUDEP-7 Inventory is a 7-category test to determine SUDEP risk in patients with epilepsy. To assess SUDEP risk, the test evaluates the following items: 1. Generalized tonic–clonic (GTC) seizure frequency >3 in last year; 2. GTC seizure frequency >0 in last year (if factor 1 present score as 0); 3. Seizure of any type >0 in last year (if factor 4 present, score as 0); 4. Seizure of any type >50 per month in last year; 5. Duration of epilepsy ≥30 years; 6. Use of three or more AEDs; 7. Intellectual disability [29]. For some analysis, SUDEP-7 scores were stratified into a high-risk subcategory (SUDEP-7 score > 5) and low SUDEP risk sub-category (score ≤ 5) based on expected highest quartile scores in the populations with refractory epilepsy.

### 2.1. Single-Photon Emission Tomography (SPECT) Acquisition

To obtain scans with appropriate early injection based on EEG ictal onset pattern, eight patients were injected on different days for ictal SPECT, five patients were injected twice while three were shot up three times. In these cases, late injections were due to limitations in determining the ictal EEG onset for consistent early injections.

Patient scans were divided into the interictal and ictal state for analysis: (1) Interictal SPECT scans were obtained at least 24 h following the latest seizure. A scan was performed to obtain a baseline, interictal cerebral blood flow (CBF) measurement. (2) Ictal SPECT scans were obtained within 40 min of a habitual seizure captured by video-EEG monitoring. Seizure onset was based on EEG ictal onset pattern, and it was described as the earliest EEG prior to clinical evidence of seizure activity. Seizure offset, on the other hand, was defined as the last EEG or clinical evidence of seizure activity. SPECT injection time was specified when the plunger of the syringe containing the radiopharmaceutical was fully depressed. Lastly, SPECT EEG electrodes were removed, and the patient was transferred to the equipment.

Images were obtained after intravenous injection of 629 MBq of 99 mTc-ECD using a dual-head gamma camera (ECAM, Siemens, Frankfurt, Germany) with low energy high-resolution parallel-hole collimator. One hundred and twenty-eight projections in a circular orbit of 360° per subject were collected in matrices of 128 × 128 pixels with a 20% energy window centered on the photopeak of 99 mTc. In addition, one hundred and twenty projections over 360° were acquired using a 20% energy window centered on 140 keV. Then, tomographic 3D reconstruction was performed using a Depth Response Ordered Subsets Expectation Maximization (DROSEM) algorithm. A total of one hundred and twenty-eight attenuation-corrected brain slices were obtained using DROSEM algorithm. Expectation maximization is an efficient algorithm for finding the Maximum likelihood estimate. Expectation Maximum iterative scheme is characterized by discretizing continuous images and transforming image reconstruction problems into solving linear equations. The iterative algorithms have advantages over the analytical reconstruction algorithms in the case of noise, a small amount of projection data, or incomplete projection data. Moreover, experimental results show that the algorithms based on the reconstructed points discretization model and its geometric symmetry structure can effectively improve the imaging speed as well as the imaging precision. This methodology is the foundation for many popular iterative methods for images reconstruction [30,31,32].

### 2.2. Structural Magnetic Resonance Imaging (MRI) Acquisition

A 3D high resolution T1-weighted anatomical image was obtained from all subjects using an MRI scanner Siemens Aera 1.5 T (Frankfurt, Germany). A MPRAGE pulse sequence covering the whole brain was used with the following parameters: 192 sagittal slices; TR/TE = 2200/2.67 ms; slice thickness = 1 mm; acquisition matrix = 256 × 256; FOV = 256 × 256 mm^2^.

### 2.3. Cerebral Perfusion Quantification Methodology

A methodology was used to quantify cerebral perfusion SPECT images implemented on MRI co-registration and regions of interest (ROIs)-based analysis using a brain atlas [23]. The methodology involved three main steps integrated into a pipeline in python: spatial pre-processing, partial volume correction, and perfusion indexes calculation.

Spatial preprocessing encompasses brain tissue extraction, MRI-SPECT co-registration, and ROIs definition. T1-weighted MRI studies were brain-extracted using FSL BET [33,34] and segmented into gray matter, white matter, or cerebrospinal fluid using FSL FAST [32]. SPECT image was masked to include all intracerebral voxels but excluding ventricular and extracerebral CSF voxels. Three-dimensional anatomical MRI, gray-matter mask, and atlas were coregistered onto individual masked SPECT with the implementation of FSL FLIRT. ROIs were predefined utilizing the anatomical atlas. The atlas was multiplied by the subject-specific gray matter mask. Furthermore, SPECT studies were adjusted for the partial volume effect following a hybrid algorithm region-based voxel-wise correction [35].

For patient analysis, the atlas was constructed by co-registering the atlas proposed by Oishi et al., which includes 74 brain structures including cortical and subcortical structures [36]. Taking into account that the seizure onset side differs in these patients, the ipsilateral and contralateral hemisphere to the seizure onset side were used to flip the MRI/SPECT scans so as to achieve comparability. The ipsilateral regions to the seizure onset zone (SOZ) were visualized in the right hemisphere in the figures shown. The ROIs were pictured with the BrainNet Viewer (http://www.nitrc.org/projects/bnv/, accessed on 25 April 2022) [37].

#### Perfusion Index Calculation

Perfusion index (PI) was calculated by dividing the mean activity (counts per voxel) in a given region of interest (ROI) by the mean activity of the rest grey matter (equation). PI indicates perfusion of each ROI in relation to the level of global perfusion, and PI values greater than unity point to hyperperfusion.
PI ROI = AROI/Aglobal

### 2.4. Statistical Analysis

The procedure for data analysis consisted of:Determination of the perfusion index for quantifying the change between the interictal and ictal state. Hence, three states were considered for the analysis: interictal, ictal, and change.
PI Change_s_ = PI Ictal_s_ − PI Interictal_s_/PI Interictal_s_
where PI Interictal_s_ represents the perfusion value in the interictal state structure.

PI Ictal_s_ represents the perfusion value in the ictal state structure.

2.Individual analysis to select the structures involved in the most hyperperfused and hypoperfused cluster in each state. A univariate k means algorithm was used with cross-validation.

To identify perfusion patterns derived from PI in the sample such as hyperperfusion and hypoperfusion regions in each state (Interictal and Ictal) as change between states, the unsupervised classification, commonly known as clustering, was performed.

K-means algorithm based on unsupervised learning was used [38]. The steps of the algorithm include: (a) determining the number of clusters K, (b) setting centroids by first shuffling the dataset, and then randomly selecting data points for the centroids replacement, (c) calculating the distance between data points and all centroids, (d) allocating each data point to the closest cluster (centroid), (e) updating the position of the centroid according to the assigned data, (f) retaining iterating until there is no change to the centroids.

At this point, the values are assigned into K clusters without any hierarchical structure by optimizing the minimum distance between points in each of the available clusters, applying Euclidean distance between data points and centroids as a distance criterion.

The highest and lowest centroid values correspond to hyperperfused and hypoperfused cortical or subcortical region, respectively.

3.Selection features by predicting the SUDEP score using the group cluster previously identified.

To select the regions related to the SUDEP score, a method of data mining that computes an automatic feature selection was used. The procedure implies: (a) calculating a score for each region by dividing the range of values in each region into k intervals, (b) using a scoring function computed from the variable ranking (step 1) and the dependent variable (SUDEP), (c) applying the Fischer’s criterion to rank variables as a regression problem, (d) selecting the best scores for the model (step 3).

The solution to feature selection does not assume any particular type or shape of relationship between the predictors and the dependent variables (classes) of interest. Instead, the method will apply a generalized “notion of relationship” while screening the predictors one by one for regression.

All analyses were completed using Statistics software (version 12) www.statsoft.com (accessed on 1 June 2019), Tulsa, OK, USA. Modules utilized for the analyses encompassed Generalized EM and k-Means Cluster Analysis, as well as Feature Selection and Variable Screening, included in the Data Mining.

### 2.5. Ethical Considerations

The procedures performed followed the rules of the Declaration of Helsinki for human research from 1975. The scientific and ethical committee of the International Center approved this study for Neurological Restoration (CIREN 45/2020).

## 3. Results

### 3.1. Demographical and Clinical Data

Nine consecutive patients with mean age 22.66 ± 7.08 years (range: 15–33), age at epilepsy onset 6.2 ± 4.02 years (range: 1 month–34 years), and mean duration of epilepsy ± 16.22 ± 8.5 years (range: 3–33). Five males (55.5%) were included in this study. Clinical and demographic features are shown in Table 1. All patients had seizures for at least two years using no less than two major antiepileptic drugs (Carbamazepine, Diphenylhydantoin, Valproate, Phenobarbital, Primidone) at the maximum tolerated doses for adequate time periods. Of the nine patients, 66.6% were receiving polytherapy. Lamotrigine, Carbamazepine, and Clobazan were the most commonly used antiepileptic treatments in these subjects.

Non-lesional MRI was detected in six out of nine patients of the study population (66.6%). Additionally, involvement of eloquent brain regions was observed in four patients (44.4%). Six patients (66.6%) were submitted to surgical treatment. The median postsurgical follow-up period was 12 to 36 months.

All of the nine subjects underwent interictal and ictal SPECT coregistered with EEG. Mean time between the earliest EEG seizure activity and injection for ictal SPECT image acquisition was 6.8 ± 4.5 s (range: 2–17), and the mean seizure duration was 91.82 ± 41.7 s (range: 19–109 s) (Table 2).

The total score of the revised SUDEP-7 ranged from 3 to 7, mean = 5.6 (SD 1.65), SE 0.55, median 7. High and low risk determination was mainly based on experience and on focal to bilateral tonic–clonic seizure (FBTCS) frequency. In general, high-risk subjects were those who had more than three FBTCS per year. Three patients had SUDEP 7 scores between 3 and 5 (Table 1).

### 3.2. Brain Perfusion Patterns in Each Subject

The Perfusion index of the 74 brain structures per patient derived from twenty-nine SPECT images were analyzed. The optimal number of clusters (K) was estimated using a modified v-fold cross-validation for the use of K means algorithm. Five clusters were extracted. The centroid values were compared to identify the two regions of interest (ROI’s) that represent the hyperperfused and hyperperfused cortical and subcortical regions (highest centroid value and lower centroid, correspondingly). The same analysis was carried out for determining the change pattern (ROIs associated with a PI increase, PI decrease or no change in the transition from the interictal to ictal state).

In the interictal state, hypoperfused structures involved in the seizure onset zone were characterized by a PI that ranged between 0.42 and 0.64, while in ictal hyperperfused structures, PI oscillated between 1.16 and 1.46. Table 2 provides a description of the perfusion index mean of the brain structures contained in the hypoperfused and hyperperfused ROIs during the interictal and ictal SPECT in each patient. A quantification of the amounts of cortical and subcortical structures is also included.

Interestingly, in common with the interictal SPECT analysis, the medulla 8/9 (88.8%), substantia nigra 7/9 (77.7%), and red nucleus 5/9, (55.9%) were the hypoperfused ROIs in the largest number of evaluated patients during the ictal SPECT.

The entorhinal area on the other hand was recognized as the hypoperfused ROIs solely in the ictal state. The globus pallidus ipsilateral to the SOZ 4/9 (44.4%) was involved in the hyperperfused ROIs during the ictal SPECT (Figure 1A,B). In the largest part of patients, the cingulate and putamen were included in the hyperperfused ROIs in both states. A PI increase in the transition from the interictal to ictal was primarily observed in the hyperperfused ROIs. A PI decrease, on the other hand, was mainly detected in the hypoperperfused ROIs in 8/9 (88.8%) patients (Figure 1C).

Summarizing, perfusion patterns in the two evaluated states indicated as hyperperfused ROIs at seizure onset the bilateral rectus gyrus, putamen, as well as globus pallidus. Hypoperfused ROIs involved subcortical structures in both cerebral hemispheres; namely, the medulla oblongata, red nucleus, substantia nigra, and entorhinal area. On the other hand, during the interictal state, hyperperfused ROIs were the bilateral rectus gyrus and putamen. Hypoperfused ROIs included subcortical structures such as the medulla oblongata, red nucleus, and substantia nigra.

There was no relationship between the injection time, the PI, and the number of clusters identified with hyper and hypoperfusion during both the interictal and ictal state. Furthermore, no differences in the number and PI of the hyperperfused and hypoperfused subcortical and cortical structures between the evaluated epilepsy types were found. As can be seen in Table 2, there was a tendency towards a greater number of the hypoperfused subcortical structures (8–12) and hyperperfused cortical structures (10–13).

No differences in PI between the interictal and ictal hyper and hypoperfusion ROIs were observed (χ^2^(1) = 0.111, *p* = 0.738), (χ^2^(1) = 0.142, *p* = 0.705) in the identified clusters. In summary, the mean PI of the structures involving in the most prominent hyperperfusion ROIs in patients evaluated during the ictal SPECT was 1.322 ± 0.114 (1.160–1.460), and 1.319 ± 0.243 (1.140–2.020) in the interictal. The structures in the ictal hypoperfusion ROIs exhibited a PI mean of 0.544 ± 0.085 (0.410–0.700) and 0.562 ± 0.094 (0.410–0.730) in the interictal.

### 3.3. Relationship between Perfusion Patterns and Sudep-7 Inventory Using Feature Selection and Variable Screening

A tendency to correlation between SUDEP 7 score and the brain structure number contained in the most significant hypoperfusion ROIs was found during seizure onset. R = 0.677, *p* < 0.05 Spearman Rank Order Correlations. Patients with the highest SUDEP score showed higher number of subcortical brain structures in the hypoperfusion ROIs during seizure onset. Patients who exhibited seven points in the SUDEP 7 inventory score revealed more than five subcortical hypoperfused structures, which included the medulla oblongata, red nucleus, substantia nigra, and entorhinal area in both hemispheres.

The feature selection and variable screening (FSL) methodology allowed identifying the change from the interictal to ictal observed in the ipsilateral putamen F = 12.60, *p* = 0.03, entorhinal area F = 25.80, *p* = 0.01, and temporal middle gyrus F = 12.60, *p* = 0.03 as a possible biomarker of SUDEP risk. In contrast to the low risk SUDEP patients, those with the highest SUDEP score showed no PI change between the interictal to ictal transition in the putamen, entorhinal area, and temporal middle gyrus (Figure 2).

## 4. Discussion

This paper revealed that reliable hypo- and hyperperfused ROIs in cortical and subcortical structures during the ictal and interictal state could be markers of severe focal epilepsy. A noteworthy result of the current study was the demonstration that minor or no perfusion change in the transition from the interictal to ictal state in the ipsilateral putamen, entorhinal cortex, and temporal middle gyrus is a potential biomarker of SUDEP risk. These structures are involved in different ways in bilateral tonic–clonic seizures (BTCS), which are considered the main risk factor related to SUDEP.

### 4.1. Main Findings

#### 4.1.1. Hypo and Hyperperfused ROIs in Cortical and Subcortical Structures during the Ictal and Interictal State in DRFE

It is interesting to note that, during the interictal and ictal state, the hyperperfused ROIs in the largest part of patients were the bilateral rectus gyrus, putamen, and globus pallidus ipsilateral to the SOZ. This finding may support previous research related to the basal ganglia implication in different types of epilepsy. Based on structural voxel-based morphometry and Granger causality analysis, other research has demonstrated deficits in grey matter volume at the caudate and putamen as well as their causality in frontal lobe epilepsy patients [39]. Earlier findings in patients with a history of insulo-opercular epilepsy using PET have also indicated that the subcortical nuclei (caudate nucleus and putamen) are regularly involved in insulo-opercular epilepsy, showing significant hypometabolism in the insular lobe, central operculum, supplementary motor area, middle cingulate cortex, bi-lateral caudate nucleus, and putamen [40]. On the other hand, in mesial temporal lobe epilepsy, the basal ganglia, insula, and anterior temporal lobe are the most commonly hyperperfused regions in SISCOM images [41]. Remarkably, reductions in D2 receptor availability in vivo have been also confirmed in patients with TLE [42].

The results in relation to hyperperfused structures during the ictal and interictal state indicate that the involvement of cortical and subcortical regions represent a connected epileptic network [23].

With respect to the hypoperfusion pattern, we found significant ROIs involved in subcortical structures such as the red nucleus, substantia nigra, and medulla during both the interictal and ictal state. It is not known if the observed hypoperfusion is the result of a primary reduction in blood flow or a consequence of reduced neuronal activity and resultant decreased perfusion due to neurovascular coupling [43].

Basal ganglia circuits are closely involved in the modulation, propagation, and cessation of seizures in both temporal and extratemporal epilepsies [44,45]. Experimental studies have also reported structural and functional changes of substantia nigra pars reticulata (SNr) neurons among different types of epileptic models [46,47]. There is also evidence from clinical practice using lesion, pharmacological interference, or deep brain stimulation targeting that the SNr can modulate the intensity of epileptic seizures [48,49,50]. The individual analysis indicated a hypoperfusion in SNr, during the interictal and ictal onset in the 67% and 77% of the patients, respectively. This corroborates the connectivity decrease in the basal ganglia, thalamus, and orbitofrontal regions using magnetoencefalography [51].

One interesting finding from this study was the hypoperfusion observed in the medulla during both the ictal and interictal state in around 70% and 90% of the patients, respectively. These results match with functional studies using MRI. In TLE patients with high SUDEP risk, two subnetworks involving the brainstem have been demonstrated: one with a decreased connectivity, and the other with an increased connectivity [13]. There is indication that tissue changes within brain structures that trigger cardiovascular and breathing collapse may be also relevant to the pathophysiology of SUDEP [1,14]. Postmortem study revealed a reduction in neuromodulatory, neuropertidergic, and monoaminergic neurons in respiratory nuclear groups in the medulla [52]. In addition, one study using the arterial spin labeling technique reported that postictal hypoperfusion in brainstem respiratory centers occurs more often following BTCS than other types of seizure. This offers a probable explanation for the increased risk of SUDEP in patients who regularly experience this kind of seizures [1]. It is coherent to theorize that long epilepsy duration especially with frequent BTCS observed in our patients may lead to progressive involvement of anatomic regions related to cardiorespiratory functions and cardiovascular that may be associated with the SUDEP risk.

The results point to the key role that brainstem structures play in the pathophysiology of SUDEP, if we take into account that this study included DRE. It has been reported that the incidence of SUDEP is much higher in these patients [53,54].

#### 4.1.2. Invariability in the Perfusion Pattern during Interictal to Ictal as Potential SUDEP Biomarkers

In the current study, the nearly invariability in the perfusion pattern during the interictal to ictal transition in the ipsilateral putamen, middle temporal gyrus, and entorhinal area was identified as a potential biomarker of SUDEP risk. There is evidence that these structures are involved in diverse ways in bilateral tonic–clonic seizures (BTCS), which represent the main risk factor related to SUDEP. The neuronal networks linked to BTCS suppress brainstem respiratory or autonomic control centers, producing hypoventilation, cardiovascular dysfunction, and SUDEP.

Our findings are in line with the single cerebral SPECT imaging study according to the literature reviewed in relation to SUDEP that has been published to date. This study demonstrated that an abnormal increased activity in subcortical structures (cerebellum, basal ganglia, brainstem, and thalamus), along with a decreased activity in the association cortex observed precisely during and following seizures may be crucial for motor manifestations and for impaired consciousness in tonic–clonic seizures [18].

As for the putamen involvement in predicting SUDEP risk stratification observed in this study, there is indication that the basal ganglia are hyperactive during FBTCS and that subjects who have FBTCS also present additional basal ganglia atrophy compared to those who do not have this seizure type [55]. So, it is highly likely that the putamen may be involved in the motor component of BTCS.

A recent study using resting state functional MRI (rsfMRI) suggest that topological architecture of the thalamocortical network together with the basal ganglia–thalamus network can provide information about the presence and the effective control of focal to BTCS [45].

The temporal cortex has been related to widespread postictal suppression in epilepsy patients with a history of BTCS. In this study, the entorhinal cortex perfusion and temporal middle gyrus were identified as significant biomarkers of SUDEP risk stratification. There are scarce findings about entorhinal implication in SUDEP. A former study indicated that the volume increase in the amygdala, entorhinal cortex, parahippocampal gyrus, and subcallosal cortex induces apnea or hypotension, and that these areas may be hyperactive. They postulated that cardiovascular and breathing collapse recovery sites may be impaired due to tissue loss, and that these structural trends may well help SUDEP risk evaluation [24,56]. This indicates that the increase of volume of entorhinal cortex may justify its hyperactivity, inducing apnea or hypotension.

Consistent with our results, a recent FDG PET study proposed a cortico-subcortical metabolic network abnormality. They also found that hypometabolism in temporal regions, mainly in the right hemisphere, has been related to widespread postictal suppression in epilepsy patients with a history of BTCS [57].

Although there are few data on specific autonomic functions associated with the temporal gyrus, parasympathetic autonomic regulation reported is relevant to SUDEP. In a study of patients with severe ictal-hypoxia during generalized seizures, lower volumes in the temporal pole and right superior temporal gyrus were detected [11]. Cortical thinning, including progressive atrophy using MRI, has been seen in superior temporal gyrus in epilepsy patients [57,58]. Concerning autonomic function, the parahippocampal gyrus in addition to other mesial temporal regions have direct connections to brainstem autonomic regulatory nuclei [59].

## 5. Conclusions

The results allowed identifying reliable hypo- and hyperperfused cortical and subcor-tical brain regions during the ictal and interictal state not directly linked to epileptogenicity, but potentially related to SUDEP risk stratification. The almost invariability in the perfusion pattern during the interictal to ictal transition in the ipsilateral putamen, entorhinal cortex, and temporal middle gyrus proved to be a potential biomarker of SUDEP risk.

## 6. Limitations

Although this research is based on a relatively small sample size of temporal and extratemporal focal epilepsies, the use of ictal and interictal SPECT images coregistered to EEG is a certain novelty value in this study. Furthermore, the application of a quantitative methodology for the analysis allowed us to detect robust common perfusion patterns unrelated to the individual seizure onset zone in DRE patients.

## Figures and Tables

**Figure 1 behavsci-12-00207-f001:**
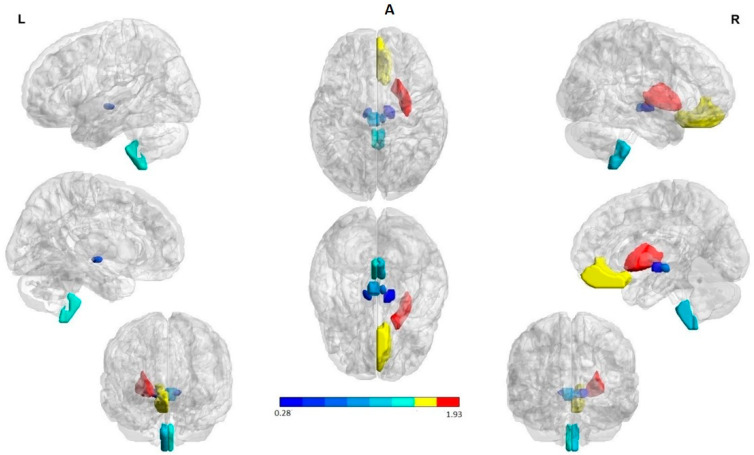
Structures with high percent match across all subjects in the hypoperfusion (in blue) and hyperperfusion (in red and yellow) ROIs. The right hemisphere to indicate the seizure onset region (ipsilateral). The first and the third column in each panel to indicate the lateral, medial, and anterior views of the left and right hemispheres, respectively. The second column (in the middle) to indicate the superior and inferior views. (**A**) Interictal state: Hyperperfused ROIs in the majority of patients were the bilateral rectus gyrus and putamen. Hypoperfused ROIs involved subcortical structures in both cerebral hemispheres; namely, the medulla oblongata, red nucleus, and substantia nigra. (**B**) Ictal state: Hyperperfused ROIs in the largest part of patients were the bilateral rectus gyrus, putamen as well as globus pallidus ipsilateral to the SOZ. Hypoperfused ROIs involved subcortical structures in both cerebral hemispheres; explicitly, the medulla oblongata, red nucleus, and substantia nigra. (**C**) State change: Structures with a significant increase and decrease perfusion between the two states across all subjects. A PI increase (in red and yellow) and a reduction (in blue) were observed in the hyperperfused and hypoperfused ROIs correspondingly. The ROIs were visualized with the BrainNet Viewer (http://www.nitrc.org/projects/bnv/, accessed on 25 April 2022) [35].

**Figure 2 behavsci-12-00207-f002:**
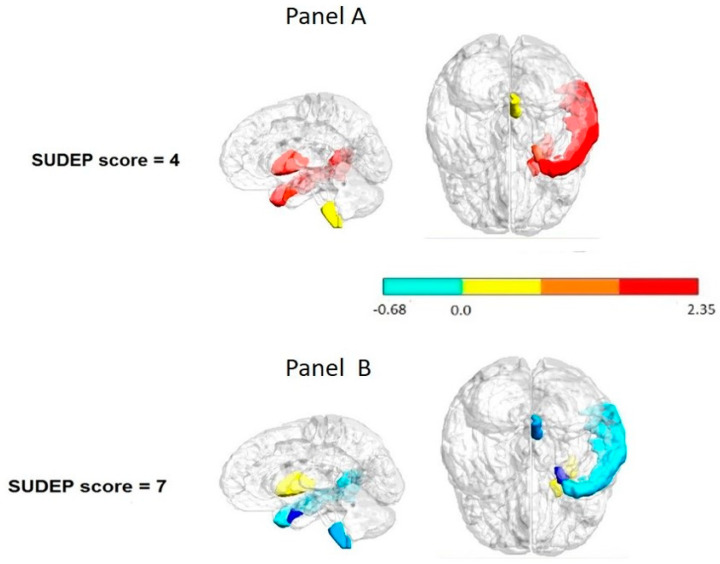
Interictal to ictal change pattern per regions considered as possible biomarkers of the SUDEP-7 risk. Panel (**A**): lower risk patient. Panel (**B**): higher risk patient. Notice that the patient with the highest SUDEP-7 score showed no change in the transition from the interictal to ictal state in the putamen entorhinal cortex and temporal middle gyrus (in blue). These structures were identified as potential biomarkers of SUDEP risk using feature selection and a variable screening methodology. The color bar to indicate the normalized scale change.

**Table 1 behavsci-12-00207-t001:** Demographic and clinical features.

Patient	Age (Years)	Age Seizure Onset (Years)	Epilepsy Duration (Years)	Sex	Epilepsy Type/Lateralization Sz Onset	Sz Duration (s)	SUDEP-7 Score	Epilepsy Surgery Type	Histopathological Findings
1	32	8.0	24	female	TLE/L	109	3	L temporal lobectomy	FCD I
2	33	0.8	32	female	TLE/L	62	5	Non operated	Non operated
3	21	6	15	male	NLFE/R	95	7	R midlle frontal gyrus topectomy plus MST	FCD I
4	25	9.0	16	female	LFE/R	73	7	Non operated	Non operated
5	15	4.0	11	male	NLFE plus Lennox Gastaut Syndrome/R	66	7	R frontal resection plus anterior callosotomy plus disconnection	Descriptive
6	15	14.0	1	female	NLFE/R	19	4	R frontal resection plus MST	FCD Ia
7	18	8.0	10	male	NLFE/L	56	7	L parietal topectomy and posterior disconnection	FCD Ia
8	17	3.0	14	male	LFE/L	35	7	anterior callosotomy plus L frontal resection	FCD I
9	28	3.0	25	male	NLFE/R	72	4	Non operated	Non operated

Sz: seizure; NLFE: non lesional focal epilepsy; LFE: lesional focal epilepsy; TLE: temporal lobe epilepsy; FCD: focal cortical dysplasia; R: right; L: left; MST: multiple subpial transection. SUDEP-7 Inventory items: 1. Generalized tonic–clonic (GTC) seizure frequency >3 in the last year; 2. GTC seizure frequency >0 in the last year (if factor 1 is present scores as 0); 3. Seizure of any type >0 in the last year (if factor 4 is present scores as 0); 4. Seizure of any type >50 per month in last year; 5. Duration of epilepsy ≥30 years; 6. Use of three or more AEDs; 7. Intellectual disability.

**Table 2 behavsci-12-00207-t002:** Perfusion index mean of the brain structures contained in the hypoperfusion and hyper-perfusion ROIs as well as in the change during the transition from the interictal to ictal state. The total of structures within each cluster and the number of subcortical regions. Patients with pairs (in-terictal and ictal) SPECT scans are shown.

Patient	INTERICTAL				ICTAL		CHANGE	
	Hypo. Cluster Mean ± SD (N/N_subc_)	Hyper. Cluster Mean ± SD (N/N_subc_)	* Time (s)	Seizure Duration (s)	Hypo. Cluster Mean ± SD (N/N_subc_)	Hyper. Cluster Mean ± SD (N/N_subc_)	Hypo. Cluster Mean ± SD (N/N_subc_)	Hyper. Cluster Mean ± SD (N/N_subc_)
1	0.51 ± 0.07 (2/1)	1.21 ± 0.02 (9/4)	7	109	0.41 ± 0.14 (3/2)	1.46 ± 0.11 (5/2)	−0.61 ± 0.09 (13/11)	0.27 ± 0.11 (11/6)
2	0.64 ± 0.02 (13/10)	1.24 ± 0.04 (4/0)	17	62	0.41 ± 0.14 (5/3)	1.29 ± 0.07 (14/9)	−0.13 ± 0.05 (19/2)	0.68 ± 0.14 (7/6)
3	0.45 ± 0.11 (7/7)	1.21 ± 0.03 (13/3)	2	95	0.46 ± 0.11 (7/6)	1.37 ± 0.00 (16/3)	−0.20 ± 0.07 (11/3)	0.37 ± 0.12 (7/2)
4	0.42 ± 0.10 (8/8)	1.16 ± 0.00 (9/1)	8	66	0.59 ± 0.15 (7/6)	1.30 ± 0.06 (4/3)	−0.12 ± 0.04 (9/1)	0.73 ± 0.44 (10/2)
5	0.57 ± 0.08 (7/7)	1.21 ± 0.04 (6/2)	4	19	0.45 ± 0.08 (4/4)	1.16 ± 0.02 (9/3)	−0.13 ± 0.03 (20/7)	0.27 ± 0.07 (4/2)
6	0.48 ± 0.02 (6/4)	1.33 ± 0.04 (6/3)	6	56	0.54 ± 0.10 (8/8)	1.28 ± 0.10 (14/3)	−0.47 ± 0.10 (2/2)	0.32 ± 0.49 (4/3)
7	0.60 ± 0.01 (3/3)	1.24 ± 0.03 (10/5)	10	35	0.50 ± 0.14 (7/6)	1.16 ± 0.04 (12/3)	−0.47 ± 0.10 (11/6)	0.11 ± 0.03 (14/3)
8	0.54 ± 0.11 (7/7)	1.16 ± 0.04 (10/1)	5	72	0.74 ± 0.03 (5/4)	1.44 ± 0.09 (4/4)	−0.11 ± 0.04 (24/6)	1.08 ± 0.21 (6/6)
9	0.62 ± 0.06 (4/3)	1.77 ± 0.20 (3/0)	3	73	0.57 ± 0.02 (4/3)	1.44 ± 0.02 (11/2)	−0.34 ± 0.11 (3/2)	0.25 ± 0.09 (15/3)

N/Nsubc: number of structures in each cluster/number of subcortical structures. Hypo: hypoperfusion, Hyper: hyperperfusion, * Time: injection time.

## Data Availability

The datasets produced for this study are available on request to the corresponding author.

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
