# Peer review of "Relation of Brain Perfusion Patterns to Sudden Unexpected Death Risk Stratification: A Study in Drug Resistant Focal Epilepsy"

_behavsci, 2022, doi:10.3390/bs12070207_

Round 1

Reviewer 1 Report

Lilia M Morales Chacón at al provide interesting results showing that brain perfusion patterns associate with the risk of sudden unexpected death in patients with refractory epilepsy. The results were obtained with single-photon emission computed tomography in 9 subjects (5 males and 4 females) at the age of 15-33 years.

I have three major questions.

First. Have the authors examined the postictal period, considering the fact that Lui at al. 2020 (https://pubmed.ncbi.nlm.nih.gov/32675079/) found a postictal hypoperfusion in brainstem respiratory centers (this paper was cited)? 

Second. From "Material and methods" (Lines 90-97), it was not clear what criteria were used to assess the risk of SUDEP. Could the authors describe these criteria in more detail?

A 7-category SUDEP test was used to determine SUDEP risk, and only two subcategories were selected, such as high-risk of SUDEP (including five patients with the score =7) and low risk (including four patients with score =3, 4 and 5). The number of patients was too low to access correlations between the risk of SUDEP and brain perfusion parameters.  Nevertheless, the authors reported a significant Spearman's correlation between SUDEP 7 score and the number of brain hypoperfused structures during seizure onset (R=0.677, p<0.05) - in lines 286-290. What structures were hyperperfused at seizure onset and what structures were hypoperfused?    

Third. This question relates to the previous one. The seizure onset zone was unilateral (Table 1 indicates - left Sz in patients 1,2, 7 and 8; right Sz in 3,4,5 and 6). Therefore, there were two sides (ipsilateral and contralateral to Sz) and two perfusion patterns (hyperperfused and hypoperfused) examined at the ictal and interictal conditions. 
In order to better understand the results, I recommend to summarize the results of perfusion patterns in two states.
THE ICTAL STATE.
- ipsilateral hyperperfused structures ...
- contralateral hypoperfused structures ...
- ipsilateral hyperperfused structures ...
- contralateral hypoperfused structures ...
PREICTAL->ICTAL STATE. 
- ipsilateral hyperperfused structures ...
- contralateral hypoperfused structures ...
- ipsilateral hyperperfused structures ...
- contralateral hypoperfused structures ...

I have a few minor comments:

  1. Lines 49-53, which refer to the results of neuroimaging studies on brain alterations in patients with high risk for SUDEP citing refs [15-17]. These references do not seem to be appropriate. Please check and revise this paragraph. 
    The same problem is in lines 56-57, referring to [22-24]. Please check.
  2. The aim of this study is not unclear. It has been stated in Lines 77-78: "evaluating interictal and ictal cerebral brain perfusion patterns in subjects with DRFE and its relation to SUDEP risk stratification." Could the authors formulate a clear hypothesis based on the literature data?
  3. Line 223. "Three patients had SUDEP 7 scores between 1 and 5. (Table 1)." In fact, these three patients had SUDEP scores between 3 and 5 (3, 4 and 5).
  4. Line 230. "that represent the hyperperfused and hyperperfused cortical and subcortical"  - maybe hypoperfused? 
  5. Overall formating of the paper needs to be corrected. Also, the list of references needs correction, inasmuch as some information is missing. 
  6. There was a common opinion that "the relationship between SPECT and surgical outcome has not been established" Devous et al., 1998 (https://pubmed.ncbi.nlm.nih.gov/9476937/). Could the authors comment on the relationship between brain perfusion patterns (based on the outcomes of SPECT analysis) and outcomes of surgical examination + EEG examination? 

Reviewer 2 Report

The reviewer has reads with pleasure the content of the manuscript, which addresses one of the most important research factors regarding relation of brain perfusion patterns to sudden unexpected  death risk stratification.

In the opinion of the reviewer, the text is valuable and could be reconsidered after major revision. Therefore, it is recommended improve  of text or/and detailed clarification response to reviewer's raised issues:

- establish conclusions on ground  of such limited sample size (n=9) of this manuscript may be questionable. Studies on a larger group of patients than nine patients  could be scientifically more valuable

- in reviewer understanding, at least two drugs were administered to only 66.6% of 9 patients, if ( assuming) 33.4% of 9 patients received only one drug. What was the reason to administer only one drug in case of SUDEP?

- please put the SUDEP criteria (0-10) in the table for a precise description

- It is necessary  detailed explanation of Depth Response  Ordered Subsets Expectation Maximization (DROSEM) algorithm and credible references in the field must be provided

- It is necessary detailed explanation of CIREN Comprehensive Epilepsy Surgery Program and credible references in the field must be provided

- in some parts of manuscript abbreviation must be used more accurately  , i.e. (PI), ROI, ROIs ROI`s

- references must be re-edited to comply with the journal guideline

Round 2

Reviewer 2 Report

Most of the comments are corrected as expected by the reviewer. References still need to be improved - this should be done in accordance with the rules in force at MDPI